# Design and Non-Linearity Optimization of a Vertical Brushless Electric Power Steering Angle Sensor

**DOI:** 10.3390/s24082469

**Published:** 2024-04-12

**Authors:** Jie Chen, Yanling Guo

**Affiliations:** College of Mechanical and Electrical Engineering, Northeast Forestry University, Harbin 150000, China; 15663712907@163.com

**Keywords:** contactless angle sensor, linearization, particle swarm optimization

## Abstract

This paper presents the design and the non-linearity optimization of a new vertical non-contact angle sensor based on the electromagnetic induction principle. The proposed sensor consists of a stator part (with one solenoidal excitation coil and three sinusoidal receiver coils) and a rotor part (with six rectangular metal sheets). The receiver coil was designed based on the differential principle, which eliminates the effect of the excitation coil on the induced voltage of the receiver coil, and essentially decouples the excitation field from the eddy current field. Moreover, the induced voltages in the three receiver coils are three-phase sinusoidal signals with a phase difference of 10°, which are linearized by CLARK transformation. To minimize the sensor non-linearity, the Plackett–Burman technique was used, which identified the stator radius and the rotor blade thickness as the key factors affecting the sensor linearity. Then, the particle swarm algorithm with decreasing inertia weights was utilized to optimize the sensor linearity. A sensor prototype was made and tested in the laboratory, where the experimental results showed that the sensor non-linearity was only 0.648% and 0.645% in the clockwise and counterclockwise directions, respectively. Notably, the non-linearity of the sensor was less than −0.696% at different speeds.

## 1. Introduction

Angle sensors are widely used in various applications where precise angle measurements are required, including automobiles [1], aviation servo systems [2], and industrial robots [3]. Depending on the underlying angle measurement method, angle sensors can be divided into contact angle sensors [4] and non-contact angle sensors [5,6,7]. Contact angle sensors are mainly the potentiometer-type angle sensors [8], where the presence of friction during measurements makes these sensors more prone to wear, electrical noise, short life, etc. Non-contact angle sensors, on the other hand, can be further divided into capacitive, photoelectric, and magnetoelectric angle sensors. Capacitive angle sensors have the advantage of low power consumption and high sensitivity [9,10,11], but this sensor has poor temperature stability and low measurement accuracy. Meanwhile, photoelectric angle sensors need an independent light source, and since environmental factors affect the light propagation, such sensors require tight sealing to ensure their use in harsh environments [12,13,14]. Magnetoelectric angle sensors include Hall sensors, magnetostrictive sensors, and electromagnetic induction-type sensors. Hall-type angle sensors have the advantages of high reliability, long life, and fast response [15,16], but this sensor contains a specific structure of the internal permanent magnet, its interchangeability is relatively poor, and the output is non-linear. In addition, the signal of Hall-type sensors is affected by temperature, thus needing a temperature compensation device, and the sensor accuracy is relatively low. Alternatively, magnetostrictive angle sensors have simple installation, high sensitivity, good stability, high power, and high overload capacity [17,18]. However, these sensors have a poor resistance to interference and cannot be used with magnetically conductive materials. In contrast, with the advantages of a good structural flexibility, low cost, no required temperature compensation, high environmental adaptability, high accuracy, and good electromagnetic compatibility [19,20,21], electromagnetic induction angle sensors are quite popular in many engineering applications. This paper mainly studies the angle sensor used in the Electric Power Steering (EPS) system. Measuring the angle is an important function of EPS sensors. The function of the EPS system is to detect the torque and direction generated by the steering wheel when the driver is steering. A critical component of the vehicle chassis system, the steering system directly influences the stability, driving comfort, and driving safety of the vehicle. The development of the steering system has progressed through four stages: (i) a Mechanical Steering system; (ii) a Hydraulic Power Steering system; (iii) an Electro-Hydraulic Power Steering; and (iv) an Electric Power Steering system. Among these developments, the EPS system has the advantage of energy saving and environmental protection, and is increasingly used in automobiles.

Existing electromagnetic induction type angle sensors have a flat structure [22,23,24,25], which mainly has the following defects: relatively large sensor size due to the planar windings; and the two rotors in the sensor exhibit a crosstalk and affect the sensor non-linearity. These issues of planar induction sensors indicate the need to design a new induction angle sensor with a small size and a high linearity. The structural parameters of the electromagnetic induction type angle sensor affect the intensity of the magnetic induction inside the sensor, which in turn affects the linearity of the relationship between the measured angle and the output signal [26]. Non-linearity is an important parameter to measure the performance of sensors [27]. To effectively reduce the non-linearity of an angle sensor, it is necessary to optimize the structural parameters of the sensor.

Usually, an optimization algorithm, such as a genetic algorithm [28], a response surface method [29], or a particle swarm algorithm [30], is used for the sensor structure optimization. The selection of genetic algorithm parameters, such as mutation rate and crossover rate, seriously affects the quality of the corresponding optimal solution. At present, the selection of these parameters mostly depends on the experience. The response surface method is not suitable for discrete optimization, and the optimal parameters obtained by this method depend on the degree of fit of the regression equation. The Particle Swarm Optimization (PSO) algorithm was first proposed by Eberhart and Kennedy in 1995, the basic concept of which originates from the study of the foraging behavior of bird flocks. This algorithm is fast and efficient in searching, does not depend on the problem information, and has a strong generality. The PSO algorithm is a generalized swarm intelligence method, commonly used for solving global optimization problems. In this paper, we design a new vertical non-contact induction angle sensor, where we use the Plackett–Burman test to screen the key factors affecting the non-linearity of the sensor, and then use the PSO algorithm to optimize these selected key factors.

## 2. Theoretical Research and Simulation Analysis

### 2.1. Theoretical Research

The proposed vertical non-contact induction angle sensor consists of a stator and a rotor, as shown in Figure 1. The stator includes the receiving coil and the excitation coil. Furthermore, the receiving coil winding consists of three sinusoidally structured coils with a sinusoidal period of 60°, where the spacing between the adjacent receiving coils is 10°. The excitation coil is a 10-turn solenoidal coil, while the rotor consists of six rotor blades, each at 30°.

The measurement principle of the proposed sensor can be considered as “electric–magnetic–electric”. When the excitation coil is fed with a high-frequency alternating current, an alternating magnetic field (${\vec{B}}_{e}$) is generated, and according to the Faraday law of electromagnetic induction, the voltage is induced in the receiving coil. The rotor in the alternating magnetic field induces a secondary magnetic field (${\vec{B}}_{r}$) of the same frequency, and the eddy current field here also induces a voltage in the receiving coil. Since the rotor is distributed at intervals, the coupling area between the rotor and the receiving coil is related to the rotor rotation angle, and the variation of the coupling area changes the induced voltage in the receiving coil. Therefore, the receiving coil induced voltage is due to the superposition of the excitation magnetic field ${\vec{B}}_{e}$ and the eddy current field ${\vec{B}}_{r}$, as shown in Equation (1). Next, the magnetic induction intensity can be calculated as shown in Equation (2).
(1)$$U=N\frac{d\varphi }{dt}=N\frac{d\left(\int {\vec{B}}_{e}(t)+{\vec{B}}_{r}(t)\right)dS}{dt}$$
(2)$${\vec{B}}_{e,r}=\frac{\mu_{0}}{4\pi }\int \frac{Id\vec{l}\times \vec{r}}{r^{2}}$$
where *N* is the number of turns; *S* is the area; *I* is the excitation current; $\vec{r}$ is the distance between the excitation source and the induced conductor; ${\vec{B}}_{e,r}$ is ${\vec{B}}_{e}$ and ${\vec{B}}_{r}$, respectively; $Id\vec{l}$ is the current element; and $\mu_{0}$ is the permeability of vacuum, $\mu_{0}$ = 4*π* × 10^−7^
*N* ● *A*^−2^.

According to the law of electromagnetic induction as given in Equation (1), the induced voltage is related to the magnetic induction intensity and the magnetic flux through a closed loop area. In this work, the receiving coil is designed as a differential structure, as shown in Figure 2a. Specifically, the geometry of the two adjacent loops of the receiving coil is the same. The induced voltages are equal in magnitude and opposite in direction to the excitation magnetic field, so the effect of the excitation coil on the induced voltage of the receiving coil is zero. Evidently, the differential structure of the receiver coil eliminates the influence of the excitation field on the induced voltage of the receiver coil. It should be noted that the coupling area between the rotor and the receiver coil is the key factor affecting the sensor-induced voltage. According to Figure 2a, for the sake of illustration, the receiving coil can be regarded as a series connection of the primary coil and the secondary coil. The coupling area between the rotor and the receiving coil changes as follows: (a) when the rotor is at position 1, the rotor is in the middle of the loop consisting of the primary and secondary coils, and the direction of the voltage induced by the primary and secondary coils is demonstrated by the arrow in the figure. The induced voltages of sinusoidal loop 1 and sinusoidal loop 2 are of the same magnitude but with opposite directions (the direction of the induced voltage in sinusoidal loop 1 is indicated by a red arrow, while that in sinusoidal loop 2 is indicated by a yellow arrow). Hence, they cancel out each other, and the induced voltage of the receiving coil is zero, i.e., the zero position in Figure 2b; (b) as the rotor rotates, the coupling area between the rotor and sinusoidal loop 1 gradually decreases, while the coupling area between the rotor and sinusoidal loop 2 gradually increases. Likewise, the magnetic flux through sinusoidal loop 2 gradually increases, and the induced voltage also gradually increases. When the rotor rotates to position 2, the rotor and sinusoidal loop 2 are fully coupled, and at this point, the magnetic flux through the sinusoidal loop 2 reaches the maximum. Meanwhile, the coupling area between the rotor and the sinusoidal loop 1 is 0, hence the magnetic flux within sinusoidal loop 1 is also zero. Here, the induced voltage of the receiving coil is at the maximum, as shown by *S*_max_ in Figure 2b; (c) as the rotor rotates further, the coupling area between the rotor and sinusoidal loop 2 gradually decreases, whereas the coupling area between the rotor and sinusoidal loop 3 gradually increases, and the induced voltage slowly decreases. When the rotor is again in the middle of the loop consisting of the primary and secondary coils, the induced voltage in the receiving coil is zero; (d) with more rotation of the rotor, the receiving coil induced voltage reaches a negative maximum, where the rotor is fully coupled to sinusoidal loop 3, corresponding to position −*S*_max_ in Figure 2b. Finally, as the rotor rotates to position 3, the receiving coil induced voltage becomes zero again, completing one whole cycle of the measurement.

According to the above analysis, the magnitude of the induced voltage of the receiving coil is related to the magnetic induction intensity and the coupling area, *S*, between the rotor and the receiving coil. The coupling area between the rotor and the receiving coil is elaborated in Figure 3, and accordingly, the variation of the rotor–receiver coil coupling area can be expressed as Equation (3).
(3)$$dS=\int_{\theta }^{\theta +\frac{\pi }{N_{R}}}\frac{2\pi r_{d}}{N_{R}}\cdot \frac{h_{j}}{2}\sin\left(\frac{\pi }{N_{R}}\theta \right)d\theta$$
where *r_d_* is the stator radius; *N_R_* is the number of rotor blades; and *h_j_* is the receiving coil height.

### 2.2. Simulation Analysis

To analyze the magnetic induction intensity, eddy current, and induced voltage of the vertical non-contact induction angle sensor, Solidworks was used to build the sensor model based on the structural parameters given in Table 1, which was then imported into COMSOL for the simulations. In the simulations, the material for the excitation coil, the receiver coil, and the rotor was set to copper, and the grid division results are shown in Figure 4.

The input voltage for the excitation coil is shown in Figure 5a, i.e., *U* = 5 × sin(2π × 10,000 × *t*) *V*, while the voltage induced in the receiving coil is shown in Figure 5b. Here, the induced voltage was generated using the rotor eddy current field, and according to the law of flutters, the direction of the induced voltage is opposite to that of the excitation voltage. The induced voltage was generated from the excitation coil to the rotor and then to the receiving coil. At the initial stages, the induced voltage had not yet reached the steady state and had a small amplitude. After the first cycle, the induced voltage reached the steady state with an amplitude of 34.5 mV and a frequency of 10 kHz, which was the same as the excitation frequency. The magnetic induction intensity of the rotor is shown in Figure 5c, where it can be seen that the magnetic induction intensity was large inside the excitation coil; its maximum is 12.6 × 10^−3^ T. The rotor vortex distribution is shown in Figure 5d, and the maximum rotor vortex density was 1.41 × 10^−7^ A/m^2^. From Figure 5d, it can be seen that the vortex was mainly distributed at the bottom and the sides of the rotor.

The receiving winding of the sensor consisted of three coils, which were staggered in a spatial layout with a 10° offset. The induced voltages in the three receiving coils represent sinusoidal curves of the same frequency and amplitude, but with a phase difference of 10°, i.e., a three-phase signal, as shown in Figure 6. This three-phase signal can be mathematically expressed using Equation (4).
(4)$$\left\{\begin{array}{l} U_{1}=U_{j}\sin\left(\omega_{e}\theta \right)\\ U_{2}=U_{j}\sin\left(\omega_{e}\theta +2\pi /3\right)\\ U_{3}=U_{j}\sin\left(\omega_{e}\theta -2\pi /3\right) \end{array}\right.$$
where, *U*_1_, *U*_2_, and *U*_3_ are the induced voltages of the receiving coil; *U_j_* is the amplitude of the induced voltage; and *ω_e_* is the angular frequency of the induced voltage.

## 3. Non-Linearity Optimization

The structural parameters of the vertical non-contact induction angle sensor affect the magnetic induction intensity and the eddy current distribution, which in turn affect the waveform of the induced voltage in the receiving coil. Notably, unreasonable structural parameters lead to distortion of the induced sine waveform, resulting in large non-linearity. Limiting the magnitude of non-linearity is critical to enable a satisfactory performance of the sensor; therefore, the structural parameters of the sensor also needed to be optimized for reduced non-linearity. Before optimization, the three induced voltages needed to be linearized. According to Equation (5), the CLARK transformation was performed on the three induced voltages to transform the three-phase signal into a two-phase signal. The signals before and after the transformation are shown in Figure 7a. As shown in Figure 7a, the signal was converted from three-phase to two-phase with the same signal amplitude. Next, the transformed two-phase signal was linearized by Equation (6); the linearized signal is shown in Figure 7b.
(5)$$\left[\begin{array}{c} C_{1}\\ C_{2} \end{array}\right]=\left[\begin{array}{c} k\sin\left(\omega_{e}\theta \right)\\ k\cos\left(\omega_{e}\theta \right) \end{array}\right]=\frac{2}{3}\left(\begin{array}{c} \begin{array}{ccc} 1 & -\frac{1}{2} & -\frac{1}{2} \end{array}\\ \begin{array}{ccc} 0 & \frac{\sqrt{3}}{2} & -\frac{\sqrt{3}}{2} \end{array} \end{array}\right)\left[\begin{array}{c} U_{1}\\ U_{2}\\ U_{3} \end{array}\right]$$
(6)$$L=ATAN2(C_{1},C_{2})$$
where *C*_1_ and *C*_2_ are the two-phase quadrature winding voltages; and *L* is the linearized signal.

The induced voltages of the three receiving coils during one cycle of the rotor rotation were simulated. Next, the induced voltages were linearized and the non-linearity was calculated according to Equation (7). The results are shown in Table 2, demonstrating that the non-linearity between the simulated and theoretical values was 0.383%.
(7)$$L=\frac{{\left(U_{i}-U_{s}\right)}_{\max}}{U_{\max}}\times 100\%$$
where $U_{i}$ is the ideal voltage; $U_{s}$ is the simulation voltage; and $U_{\max}$ is the voltage range.

By analyzing the results in Table 2, it was concluded that the receiving coil induction voltage determined the non-linearity of the vertical non-contact induction angle sensor. Combining Equations (1)–(3), the most intuitive parameters characterizing the influence of the excitation magnetic field and the eddy current magnetic field on the induced voltage of the receiving coil include: the number of turns of the excitation coil and the excitation coil wire width (to characterize the excitation magnetic field effect); the rotor blade thickness and height (to characterize the vortex field action); the air gap between the stator and the rotor (to characterize the joint action of two magnetic fields); and the height of the receiving coil (to characterize the effect of the magnetic field lines passing through the area of receiving coil, generated by the two magnetic fields). The air gap between the stator and the rotor was determined by the stator radius and the rotor radius. In this work, the rotor radius was fixed at 14 mm; therefore, the change in the stator radius mainly characterized the air gap between the stator and the rotor.

Simultaneous multi-objective optimization often results in wasted resources and low efficiency; therefore, the key factors affecting the sensor non-linearity index need to be appropriately identified before the actual optimization. The Plackett–Burman test is a test method for analytical analysis that is used to estimate the effect of a factor as accurately as possible with a minimum number of trials. This method is suitable for quickly and efficiently screening the most important factors among all the influencing factors, which can be then further investigated. The experimental process here was to code multiple factors at high and low levels, analyze the inter-subjective effects and significance levels of the factors based on the results of Plackett–Burman test, and then filter out the factors with the highest effect on the test results. This essentially reduced the number of factors involved in the optimization. The non-linearity of the vertical non-contact induction angle sensor was first evaluated in terms of the number of turns of the excitation coil, the stator radius, the excitation coil line width, the rotor blade thickness, the rotor length, and the receiver coil height, and then the main affecting factors were selected. Specifically, the above six factors were coded and the lowest (−1) and highest (1) levels were taken for each factor for 12 sets of tests. The experimental factor codes and the high and low levels taken are shown in Table 3.

Progressively, the Plackett–Burman experimental design scheme is shown in Table 4, while Table 5 shows the evaluation table for the effect of each factor. Next, the data were subjected to a multi-distance regression analysis and the optimal equation for the non-linearity was obtained, as shown in Equation (8).
(8)$$\begin{array}{l} L=0.43+1.667\times 10^{-3}X_{1}+0.027X_{2}-2.333\times 10^{-3}X_{3}\\ +0.012X_{4}+8.333\times 10^{-4}X_{5}+1.5\times 10^{-3}X_{6} \end{array}$$

By analyzing the evaluation table for the effect of each factor (Table 5), the key factors affecting the sensor non-linearity were sorted according to the magnitude of their effect, as follows: *X*_2_ > *X*_4_ > *X*_1_ > *X*_6_ > *X*_5_ > *X*_3_. Among them, the effects of the stator radius and the rotor blade thickness reached an extremely significant level (*p* < 0.001), while the effects of the excitation coil turns, the excitation coil wire width, the rotor length, and the receiver coil height were not very significant. Therefore, two key factors, the stator radius and the rotor blade thickness, were selected for inclusion in the subsequent optimization.

Typically, the factors screened using the Plackett–Burman test are used in response surface analysis studies. The optimal parameters obtained using the response surface optimization method depend on the degree of fit of the regression equation, where different regression equations yield different optimal values. In contrast, this paper used the PSO algorithm to optimize the screened factors for reduced non-linearity, which is a mainstream optimization algorithm with fast convergence and only a few setup parameters for the sensor.

In the PSO algorithm, the PSO search space is an n-dimensional space, where the particle population consists of *N* particles and each particle in the population is initialized with a randomized position (${\vec{x}}_{i}$) and velocity (${\vec{v}}_{i}$), i.e., ${\vec{x}}_{i}^{\left(t\right)}=(x_{i,1}^{\left(t\right)},x_{i,2}^{\left(t\right)},\cdot \cdot \cdot ,x_{i,n}^{\left(t\right)})$; ${\vec{v}}_{i}^{\left(t\right)}=(v_{i,1}^{\left(t\right)},v_{i,2}^{\left(t\right)},\cdot \cdot \cdot ,v_{i,n}^{\left(t\right)})$. At time, *t*, the position, ${\vec{x}}_{i,t}$, can be considered as a set of coordinates of a point in the n-dimensional space. The particles fly through the virtual space looking for the candidate solutions and attracting the surrounding particles to a location that produces the best results. Moreover, ${\vec{p}}_{i}^{\left(t\right)}=(p_{i,1}^{\left(t\right)},p_{i,2}^{\left(t\right)},\cdot \cdot \cdot ,p_{i,n}^{\left(t\right)})$ is the individual best position of the *i*-th particle at moment, *t*, and ${\vec{g}}_{i}^{\left(t\right)}=(g_{1}^{\left(t\right)},g_{2}^{\left(t\right)},\cdot \cdot \cdot ,g_{n}^{\left(t\right)})$ is the global best position of the entire particle population at moment, t. The velocity update formula of a particle in the PSO algorithm is given in Equation (9). Meanwhile, the maximum distance traveled by a particle in one iteration cycle was determined using the velocity, as expressed in Equation (10). The position and the velocity of each particle were updated once for each iteration of the algorithm. It should be noted that a larger value of inertia weight, *ω*, is beneficial for improving the global search capability of the algorithm, while a smaller value improves the accuracy of local search. Therefore, the inertia weight, *ω*, used in this work decreased linearly with the number of iterations, as shown in Equation (11), which ensured the global search capability of the algorithm while avoiding falling into the local optimal solutions.
(9)$$v_{i,j}^{\left(t+1\right)}=\omega v_{i,j}^{\left(t\right)}+c_{1-PSO}r_{1-PSO}(p_{i,j}^{\left(t\right)}-x_{i,j}^{\left(t\right)})+c_{2-PSO}r_{2-PSO}(g_{i,j}^{\left(t\right)}-x_{i,j}^{\left(t\right)})$$
(10)$$x_{i,j}^{\left(t+1\right)}=x_{i,j}^{\left(t\right)}+v_{i,j}^{\left(t\right)}$$
(11)$$\omega =\omega_{\max}\frac{t\cdot (\omega_{\max}-\omega_{\min})}{t_{\max}}$$
where *c*_1-*PSO*_ and *c*_2-*PSO*_ are the acceleration coefficients; *r*_1-*PSO*_ and *r*_2-*PSO*_ are the uniformly distributed random numbers on the interval [0, 1]; *i* = 1, 2, … *N*_PSO_, *N*_PSO_ is the particle swarm size; *j* = 1, 2,…*n*, *n* is the number of spatial dimensions; *ω* is the value of the inertia weights; *ω*_max_ is the maximum value of the inertia weight; *ω*_min_ is the minimum value of the inertia weights; and *t*_max_ is the maximum number of iteration steps.

To minimize the non-linearity of the proposed sensor, the first step was to establish the objective function of the optimization problem. The objective function needs to satisfy one or more conditions to obtain the optimal solution by adjusting the system parameters. The non-linearity optimization of the vertical non-contact induction angle sensors is a bi-objective optimization problem, and there is incommensurability between the two objectives, i.e., there exists a conflicting relationship between the two objectives. Specifically, when one of the objective functions reaches the optimality, the other objective function may become worse. On the other hand, this phenomenon is not present in single-objective optimization problems; therefore, the bi-objective optimization problem needs to be converted into a single-objective optimization problem. Accordingly, in this paper, the objective function for optimization was reconstructed, where the weight coefficients were introduced to reflect the importance of the two objectives in the overall objective function, as shown in Equations (12)–(14).
(12)$$L\left(r\right)=\frac{{\left(U_{ir}-U_{sr}\right)}_{\max}}{U_{r\max}}\times 100\%$$
(13)$$L\left(d\right)=\frac{{\left(U_{id}-U_{sd}\right)}_{\max}}{U_{d\max}}\times 100\%$$
(14)$$L_{\min}=\min[w_{r}L(r)+w_{d}L(d)]$$
where $L\left(r\right)$ is the objective function of the rotor thickness parameter; $L\left(d\right)$ is the objective function of stator radius parameter; $L_{\min}$ is the non-linear degree of reconstruction; $w_{r}$ is the rotor thickness weighting factor; $w_{d}$ is the stator radius weighting factor; $U_{ir}$ is the ideal voltage for rotor thickness; $U_{sr}$ is the simulation voltage for rotor thickness; $U_{id}$ is the ideal voltage for stator radius; $U_{sd}$ is the simulation voltage for stator radius; *U_r_*_max_ is the maximum voltage range of rotor thickness; *U_d_*_max_ is the maximum voltage range of stator radius.

$w_{r}$ and $w_{d}$ are the weight coefficients introduced to transform the bi-objective optimization problem into a single-objective extremum problem. The weight coefficients were normalized according to the effect values obtained using the Plackett–Burman test. In particular, the stator radius effect value (*E*_d_) of 0.053 and the rotor blade thickness effect value (*E*_r_) of 0.024 in Table 5 were normalized to within the range [0, 1]. Then, the weighting coefficients for the stator radius and the rotor blade thickness were expressed as:(15)$$w_{d}=\frac{E_{d}}{E_{d}+E_{r}}=\frac{0.053}{0.053+0.024}=0.689$$
(16)$$w_{r}=\frac{E_{r}}{E_{d}+E_{r}}=\frac{0.024}{0.053+0.024}=0.311$$

Additionally, the search ranges for the rotor thickness and the stator radius are shown in Table 6, and the parameters of the PSO algorithm were set as follows: the number of particles was set to 30; the learning factor *c*_1_ = *c*_2_ = 1.4945; the maximum velocity was limited to 0.05; the minimum velocity was limited to −0.05; the number of iterations was set to 30; the maximum inertia weight value *ω*_max_ = 0.9; the minimum inertia weight value *ω*_min_ = 0.4.

The relationship between the non-linearity and the number of iterations is illustrated in Figure 8, where it can be seen that the non-linearity stabilizes at 0.366% when the number of iterations exceeds 18. Progressively, as shown in Figure 9, the particle population aggregated at a non-linearity of 0.366%, where the corresponding rotor blade thickness was 0.52 mm and the stator radius was 15.1 mm.

By optimizing the rotor blade thickness and the stator radius, the optimal design parameters of the sensor were obtained, which are shown in Table 7. The simulation yielded a non-linearity of 0.366% for the vertical non-contact angle sensor, as shown in Figure 10.

## 4. Fabrication and Testing of the Sensor

In this work, a customized sensor performance test bench was used as an experimental platform to evaluate the performance of the proposed vertical non-contact induction angle sensor, as shown in Figure 11a. The main components of the experimental equipment included a servo motor system, an angle encoder, a vertical non-contact induction angle sensor, and a data acquisition system. The vertical non-contact angle sensor was mounted on a rotating shaft, as shown in Figure 11b, and Figure 11c shows the experimental data acquisition module.

The servo motor in the system was controlled to rotate at 5 r/min in the counterclockwise and clockwise directions, respectively, and the angle measured by the vertical non-contact angle sensor was collected to calculate the non-linearity of the sensor, as shown in Figure 12. The experimental results were: non-linearity was 0.648% for the clockwise sensors; non-linearity of the counterclockwise direction sensor was 0.646%.

Furthermore, to test the non-linearity of the proposed angle sensor at different rotational speeds, the servo motor was controlled to rotate at 15 r/min, 30 r/min, and 60 r/min in the clockwise and counterclockwise directions, and the corresponding results are shown in Figure 13. Moreover, the maximum values of the non-linearities are given in Table 8. From Table 8, it can be seen that the non-linearity of the vertical non-contact angle sensor was 0.696% at the maximum for the different rotational speeds. It can be concluded that the rotational speed had little effect on the non-linearity of the sensor.

## 5. Discussions

### 5.1. Sine Fit

The vertical non-contact angle sensor designed in this paper used a receiving coil with a sinusoidal structure, which provided a better sinusoidal fit than the existing diamond-shaped receiving coils. To elaborate further, suppose the diagonal of the diamond-shaped receiving coil is 1 mm, the height of the sinusoidal receiving coil is 1 mm, and the rotor blade moves at a speed of 0.1 mm/s, as shown in Figure 14a. The variation curves of the coupling area between the rotor and the diamond-shaped coil and sinusoidal coil, respectively, are shown in Figure 14b.

The curves for the variation in the rotor and diamond coil coupling area and the variation in the rotor and sinusoidal coil coupling area were fitted as the sinusoidal functions, where the degree of fit was measured using the goodness-of-fit function, *R-squared*, which can be calculated using Equation (17).
(17)$$R-squared=1-\frac{SSE}{SST}$$
where *SST* is the square sum of the difference between the original data and the average value; and *SSE* is the sum square of errors of the fitting data and the original data.

The *R-squared* interval ranges from [0, 1], and the closer its value is to 1, the better the fit. The fit of the diamond-shaped coil was calculated as 0.998, while the fit of the sinusoidal coil was 1. Experiments have shown that the sinusoidal coil has a better sinusoidal fit. From the perspective of manufacturing processes, the diamond-shaped coils are easier to implement and save manufacturing costs. However, the sinusoidal coils offer the following advantages: (1) sinusoidal coils can reduce electromagnetic interference and crosstalk in high-frequency circuits, thus improving electromagnetic compatibility; (2) for high-speed signal lines and specific protocol signal lines, sinusoidal coils can reduce signal waveform distortion and delay distortion, thereby enhancing signal integrity; (3) sinusoidal coils can avoid the “sharp corners” introduced by right-angle traces, thus reducing the complexity of the PCB layout and the volume and weight of the circuit board.

### 5.2. Magnetic Induction Intensity

Notably, the vertical non-contact angle sensor designed in this paper used the excitation coil with a solenoidal structure to generate a large magnetic induction intensity. Table 9 compares the magnetic induction intensity of the planar type angle sensor and the sensor designed in this paper. Evident from Table 9, the sensor designed in this paper produced a greater magnetic induction intensity with a smaller excitation source; therefore, the sensor designed in this paper had a stronger anti-electromagnetic interference capability.

### 5.3. Volume

The proposed sensor was compared with the HELLA Torque Angle sensor. The size of the proposed vertical non-contact angle sensor was 35 × 35 × 40 mm^3^, whereas the HELLA Torque Angle Sensor had the dimensions of 70 × 70 × 15 mm^3^. Evidently, the sensor designed in this paper offered a 50% reduction in radial size and a 32.8% reduction in overall size, compared with the HELLA torque angle sensor shown in Figure 15. This essentially makes the vertical non-contact angle sensor more suitable for installation in spaces such as vehicles or robots.

Precautions for use and safety: (1) Please operate within a temperature range of −40 to 80 °C; (2) Use the power supply voltage and load within specified ranges and specifications; (3) Keep the sensor away from other electrical equipment as far as possible to minimize the interference signals.

## 6. Conclusions

In this paper, a new vertical non-contact induction angle sensor was designed, consisting of a solenoidal excitation coil and three receiver coils that form the stator of the sensor. In addition, a ring-shaped metal sheet with six rectangular blades was utilized to form the rotor of the sensor. Using the principle of differential measurement, the receiver coil was designed to have a sinusoidal structure, and the induced voltages generated by the excitation source in the two adjacent loops of the receiver coil cancelled out each other, which decoupled the excitation field from the eddy current field. In addition, the rotor angle was measured using the change in the coupling area between the rotor and the receiver coil, which improved the sinusoidal fit of the induced voltage. To study the sensor theoretically, a finite element analysis was performed using the COMSOL 6.1 software, where the induced voltage was a sinusoidal signal with frequency of 10 kHz and an amplitude of 34.5 mV. To analyze the non-linearity of the sensor, an algorithm using the CLARK transform method with an inverse tangent function was proposed that linearized the three-phase signal, which in turn reduced the non-linearity compared with the case when the segmented lines were used. The simulation results showed that the non-linearity of proposed sensor was only 0.383%. To reduce the sensor non-linearity, the stator radius and the rotor blade thickness were identified as the key factors affecting the sensor non-linearity, using the Plackett–Burman test. Then, based on the effect values of the stator radius and the rotor blade thickness, an optimization objective function was constructed, following which the PSO algorithm with decreasing inertia weights was used to optimize the sensor non-linearity. The optimization result showed that the sensor non-linearity was minimum at 0.366%, when the rotor thickness was 0.52 mm and the stator radius was 15.1 mm. Besides, the experimental results showed that the non-linearity was 0.648% for the clockwise rotation and 0.646% for the counterclockwise rotation. The proposed vertical non-contact induction angle sensors offer the following advantages: (a) comparative studies with commercially available HELLA sensors show that the sensors in this paper were smaller in size; (b) the magnetic induction intensity of the vertical induction sensor was greater than that of the planar type induction angle sensor; (c) compared with the optical sensors and the HALL sensors, the vertical angle sensors had negligible sensitivity to moisture, dust, oil, etc. The above advantages make the proposed sensor a promising device for automotive and robotics applications.

## Figures and Tables

**Figure 1 sensors-24-02469-f001:**
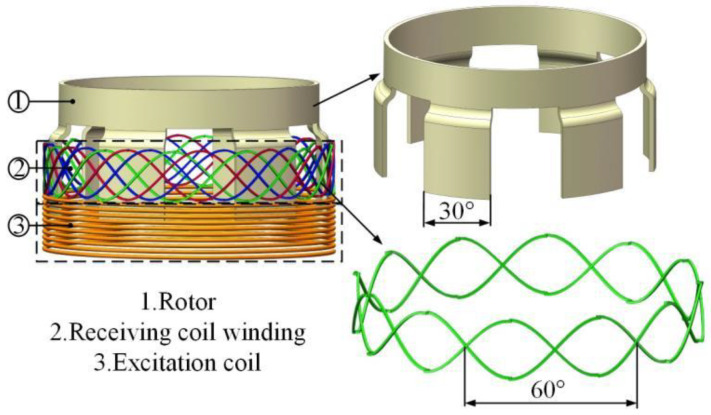
Structure of the vertical non-contact angle sensor.

**Figure 2 sensors-24-02469-f002:**
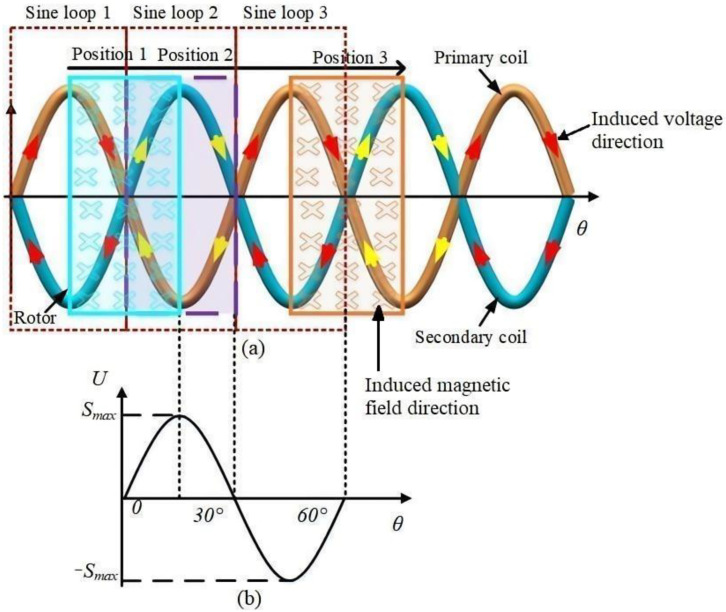
(**a**) Coupling between the rotor and receiving coil areas; (**b**) Induced voltage of the receiving coil corresponding to the different rotor positions.

**Figure 3 sensors-24-02469-f003:**
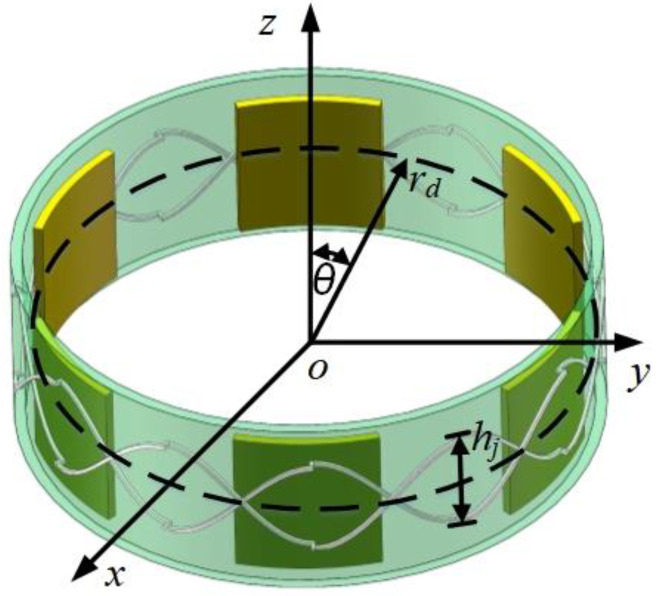
Schematic diagram of the coupling area between the rotor and the receiving coil.

**Figure 4 sensors-24-02469-f004:**
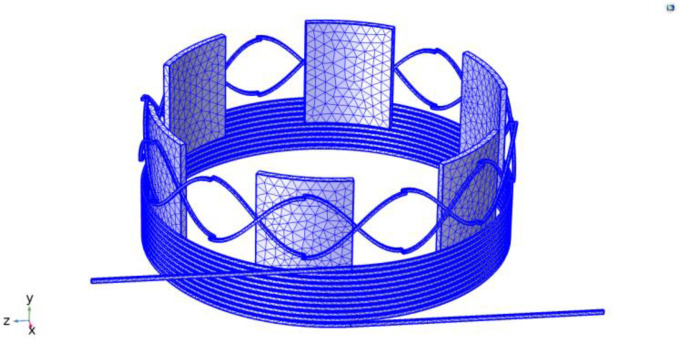
Meshing of the vertical non-contact angle sensor.

**Figure 5 sensors-24-02469-f005:**
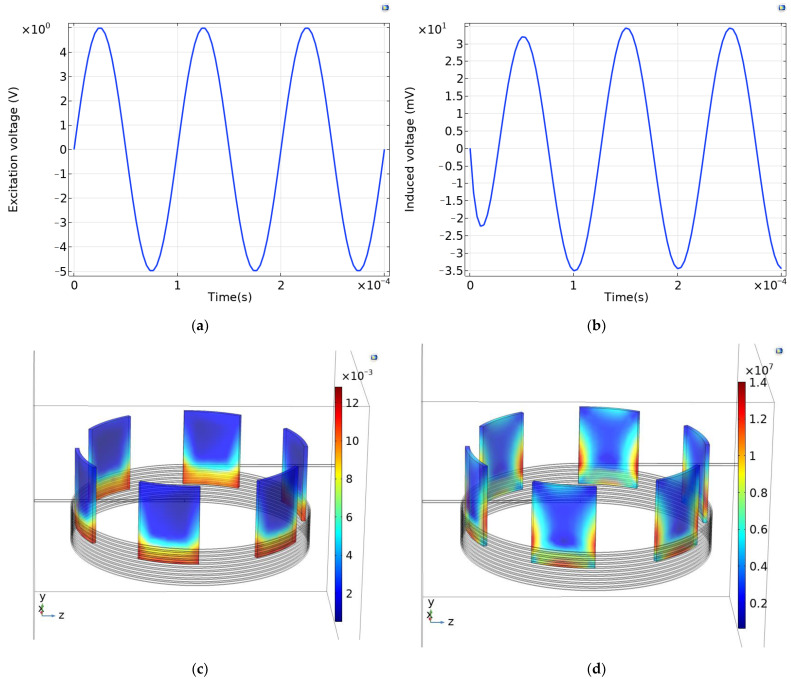
(**a**) Excitation voltage; (**b**) Induced voltage; (**c**) Rotor magnetic induction intensity; and (**d**) Rotor eddy current.

**Figure 6 sensors-24-02469-f006:**
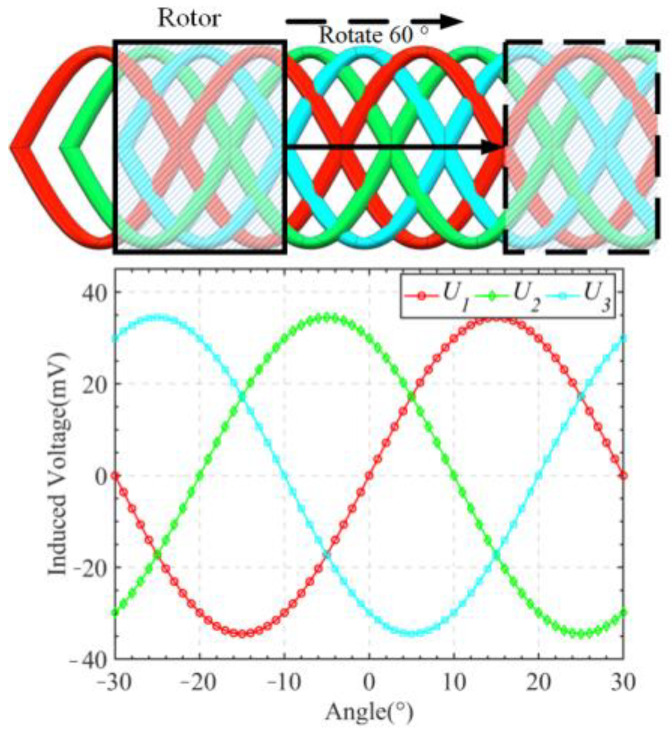
Receiving coil winding and three-phase signal.

**Figure 7 sensors-24-02469-f007:**
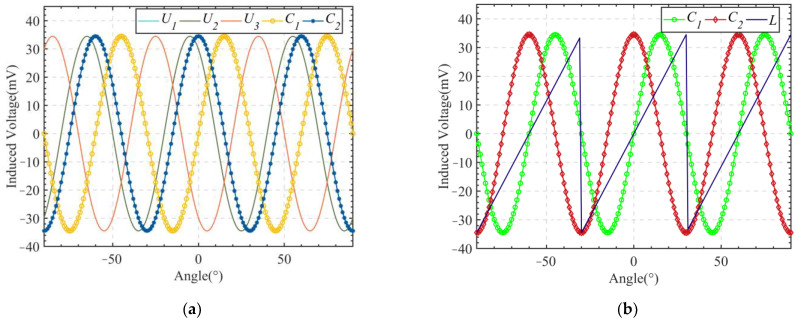
(**a**) A three-phase signal is converted to a two-phase signal; (**b**) Two-phase signal linearization.

**Figure 8 sensors-24-02469-f008:**
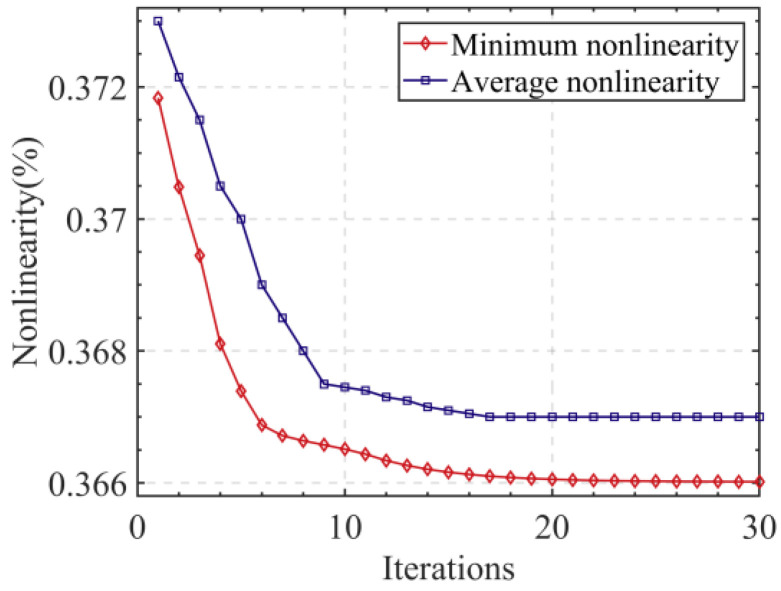
Relation between the non-linearity and the number of iterations.

**Figure 9 sensors-24-02469-f009:**
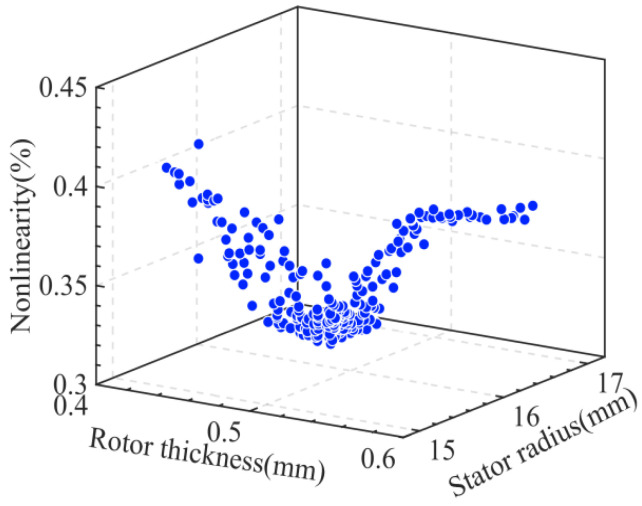
Particle motion process.

**Figure 10 sensors-24-02469-f010:**
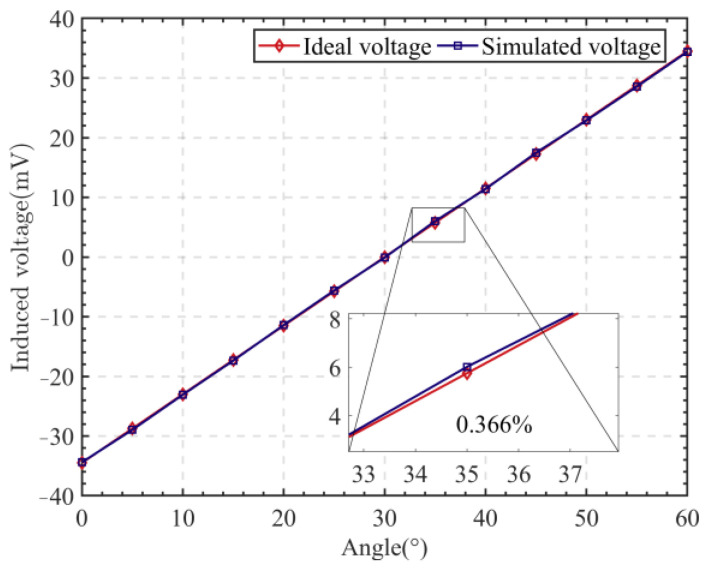
Non-linearity of the vertical angle sensor under optimal structural parameters.

**Figure 11 sensors-24-02469-f011:**
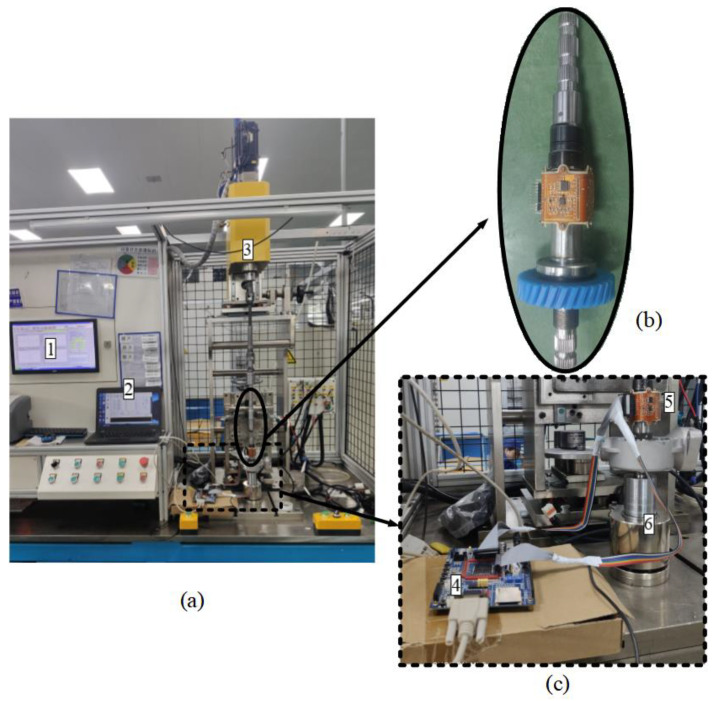
(**a**) Experimental equipment for the vertical angle sensor; (**b**) Installed angle sensor; (**c**) Data acquisition module. 1. Servo motor upper computer; 2. Upper computer of the vertical inductive angle sensor; 3. Servo motor; 4. Data acquisition card; 5. Vertical non-contact angle sensor; 6. Angle encoder.

**Figure 12 sensors-24-02469-f012:**
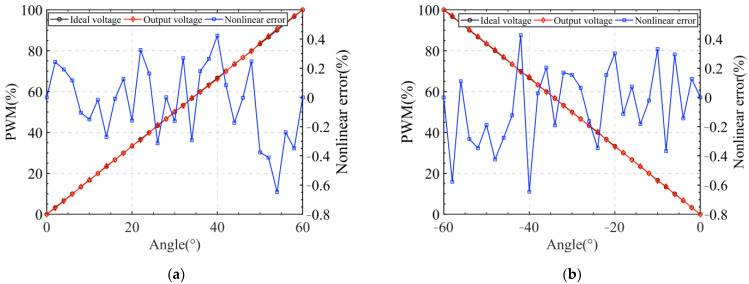
(**a**) Clockwise rotation; (**b**) Counterclockwise rotation.

**Figure 13 sensors-24-02469-f013:**
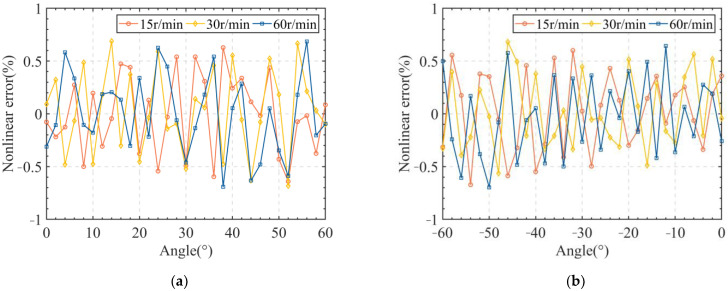
(**a**) Clockwise rotation; (**b**) Counterclockwise rotation.

**Figure 14 sensors-24-02469-f014:**
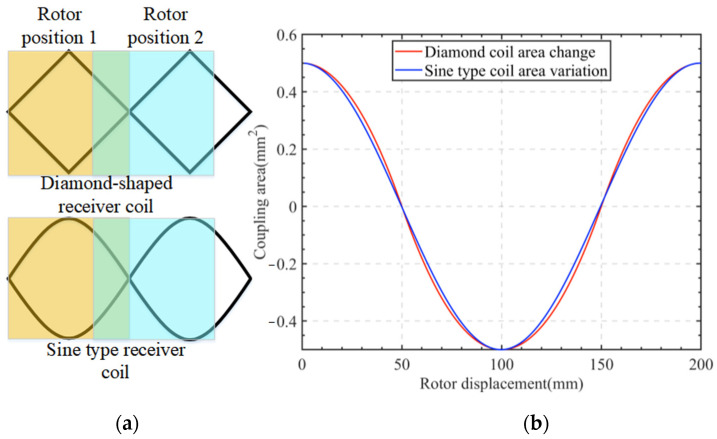
(**a**) Schematic diagram of the coupling process between the rotor and the receiving coil; (**b**) Change of coupling area between the rotor and the receiving coils.

**Figure 15 sensors-24-02469-f015:**
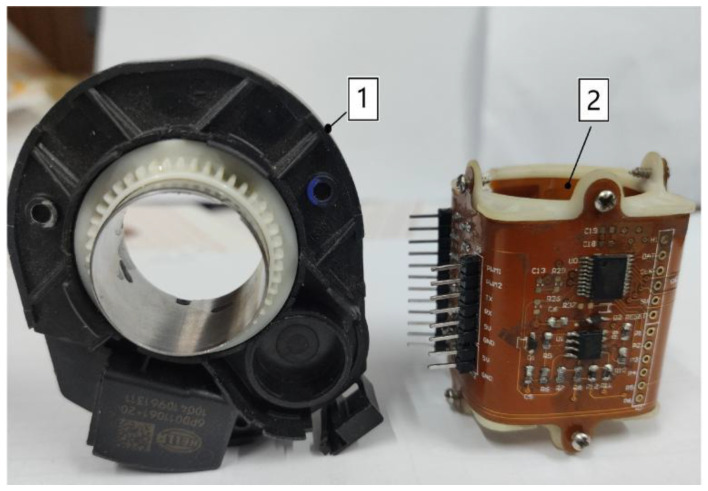
Comparison of the HELLA angle sensor and the vertical non-contact angle sensor. 1. HELLA angle sensor; 2. Vertical non-contact angle sensor.

**Table 1 sensors-24-02469-t001:** Representative structure parameters.

Structure Parameter Name	Parameter Value
Number of turns of excitation coil	10
Radius of excitation coil (mm)	14.8
Excitation coil height (mm)	6
Excitation coil line width (mm)	0.2
Rotor radius (mm)	14
Rotor height (mm)	8
Rotor blade thickness (mm)	0.5
Receiving coil height (mm)	6.5

**Table 2 sensors-24-02469-t002:** Non-linearity of vertical non-contact angle sensor.

*θ*(°)	*U_i_*(mV)	*U*_1_(mV)	*U*_2_(mV)	*U*_3_(mV)	*U*_s_(mV)	*L*(%)
0	−34.497	−0.194	−29.755	30.301	−34.399	0.143
5	−28.748	−16.699	−17.820	33.802	−28.957	0.303
10	−22.998	−29.986	−0.292	30.193	−23.082	0.121
15	−17.248	−34.182	17.051	17.655	−17.360	0.162
20	−11.499	−30.173	30.169	−0.480	−11.399	0.146
25	−5.749	−17.106	34.230	−17.861	−5.611	0.202
30	0	−0.121	30.222	−29.489	−0.104	0.150
35	5.749	18.002	16.545	−35.112	6.014	0.383
40	11.499	30.206	0.274	−30.691	11.392	0.156
45	17.248	34.351	−17.579	−16.391	17.469	0.319
50	22.998	29.439	−29.306	−0.438	22.890	0.158
55	28.748	17.626	−33.857	16.622	28.561	0.272
60	34.497	0.516	−30.067	30.089	34.391	0.154

*θ*—Target angle; *U*_1_—Induced voltage of receiving coil 1; *U*_2_—Induced voltage of receiving coil 2; *U*_3_—Induced voltage of receiving coil 3; *U_i_*—Ideal voltage; *U_s_*—Simulation voltage; *L*—Non-linearity.

**Table 3 sensors-24-02469-t003:** Non-linearity of vertical non-contact angle sensor.

Factors	Code	Low Level	High Level
Number of turns of the excitation coil	*X* _1_	8	12
Stator radius (mm)	*X* _2_	14.8	16.8
Excitation coil line width (mm)	*X* _3_	0.2	0.3
Rotor blade thickness (mm)	*X* _4_	0.4	0.6
Rotor length (mm)	*X* _5_	7.5	8.5
Receiving coil height (mm)	*X* _6_	5	6

**Table 4 sensors-24-02469-t004:** Plackett–Burman experimental design and non-linearity.

No.	*X* _1_	*X* _2_	*X* _3_	*X* _4_	*X* _5_	*X* _6_	*L* (%)
1	−1	−1	−1	−1	−1	−1	0.392
2	−1	1	1	−1	1	1	0.438
3	1	−1	−1	−1	1	−1	0.388
4	1	1	1	−1	−1	−1	0.446
5	1	1	−1	−1	−1	1	0.446
6	−1	1	1	1	−1	−1	0.459
7	1	−1	1	1	1	−1	0.415
8	1	−1	1	1	−1	1	0.411
9	−1	−1	−1	1	−1	1	0.417
10	−1	−1	1	−1	1	1	0.393
11	1	1	−1	1	1	1	0.480

**Table 5 sensors-24-02469-t005:** Plackett–Burman factor effect evaluation.

Factors	Effect Value	Coefficient	Coefficient Standard Deviation	*p*-Value	Significance
Constants		0.430	1.678 × 10^−3^	0	
*X* _1_	3.333 × 10^−3^	1.667 × 10^−3^	1.678 × 10^−3^	0.3663	
*X* _2_	0.053	0.027	1.678 × 10^−3^	0.0001	**
*X* _3_	−4.667 × 10^−3^	−2.333 × 10^−3^	1.678 × 10^−3^	0.2231	
*X* _4_	0.024	0.012	1.678 × 10^−3^	0.0008	**
*X* _5_	1.667 × 10^−3^	8.333 × 10^−4^	1.678 × 10^−3^	0.6406	
*X* _6_	3.000 × 10^−3^	1.50 × 10^−3^	1.678 × 10^−3^	0.4124	

** Indicates significant difference (*p* < 0.01).

**Table 6 sensors-24-02469-t006:** Particle search range.

Parameters	Scope
Rotor thickness	0.4 mm–0.6 mm
Stator radius	14.8 mm–16.8 mm

**Table 7 sensors-24-02469-t007:** Optimized sensor manufacturing parameters.

Structure Parameter Name	Parameter Value
Number of turns of the excitation coil	10
Radius of excitation coil (mm)	15.1 mm
Excitation coil height (mm)	6
Excitation coil line width (mm)	0.2 mm
Rotor radius (mm)	14 mm
Rotor height (mm)	8 mm
Rotor blade thickness (mm)	0.52 mm
Receiving coil height (mm)	6.5 mm

**Table 8 sensors-24-02469-t008:** Non-linearity of the sensors at different speeds.

Rotational Speed	Clockwise Direction	Counterclockwise
15 r/min	0.667%	−0.672%
30 r/min	−0.683%	0.681%
60 r/min	−0.692%	−0.696%

**Table 9 sensors-24-02469-t009:** Comparison of the magnetic induction intensity of the planar type angle sensor and the sensor designed in this paper.

Parameters	Planar Type Angle Sensor (Ref. [29])	This Work
Source of motivation	10 × sin(2π × 1,000,000 × *t*)	5 × sin(2π × 10,000 × *t*)
Magnetic induction strength	1.05 × 10^−3^ T	12.6 × 10^−3^ T

## Data Availability

The data presented in this study are available from the corresponding author upon request. The data are not publicly available due to privacy restrictions.

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
