# Peer review of "Design and Non-Linearity Optimization of a Vertical Brushless Electric Power Steering Angle Sensor"

_sensors, 2024, doi:10.3390/s24082469_

Round 1

Reviewer 1 Report

Comments and Suggestions for Authors

The paper discusses the design and optimization of a vertical non-contact EPS angle sensor based on the "electric-magnetic-electric" principle. The sensor operates by inducing voltage in the receiving coil through an alternating magnetic field generated by the excitation coil. To improve accuracy, the study focuses on minimizing sensor non-linearity. The authors also emphasize the importance of identifying and optimizing key factors to enhance sensor performance. To my only concern, it is best to provide a brief background of EPS to improve the comprehensibility of readers.

Reviewer 2 Report

Comments and Suggestions for Authors

The paper discussed the design and non-linearity optimization of a new vertical non-contact angle sensor based on the electromagnetic induction principle. It is very interesting and novel research. The paper is generally written well. The design is verified experimentally.  The receiver coil eliminates the effect of excitation coil and eddy current field. Some suggestions to improve the paper before publication

1.  Why excitation coil has 10-turns? Justify. 

2. does need to keep rotor blades in even number?

3. Declare all variables in Eq. (2)

4.  Recheck Math symbol ATAN in (6)

5. PSO algorithm has same variable as sensor equations, Say r_1, N

6.  how the value of Wr and Wd in eq. (14) chosen for objective function?

7.  Give limitations or precautions while using this sensor.

Comments on the Quality of English Language

Overall English is good in the paper.

Reviewer 3 Report

Comments and Suggestions for Authors

The paper looks interesting. I have some comments which should be taken into consideration prior to the acceptance.

1.      Remove word “new” from the title as it doesn’t carry any information.

2.      Do not use abbreviations in the title.

3.      I suggest to use “brushless” instead of “noncontact” in the title.

4.      Please check to write “Clarke” everywhere though the paper instead of “CLARKE” or “CLARK”: https://en.wikipedia.org/wiki/Edith_Clarke

5.      Figure 14 shows the difference between diamond-shaped and sine-type coils. There are several things to clarify. Please compare these shapes of the winding in terms of manufacturing process. Which is easier to manufacture, assemble, and so on? What are the voltage spectrums for both solutions? I can see that third harmonic is bigger for diamond shape; however, its impact in case of three-phase measurement system is not high. Please provide this information.
